# Key Factor Study for Generic Long-Acting PLGA Microspheres Based on a Reverse Engineering of Vivitrol^®^

**DOI:** 10.3390/molecules26051247

**Published:** 2021-02-25

**Authors:** Yabing Hua, Zengming Wang, Dan Wang, Xiaoming Lin, Boshi Liu, Hui Zhang, Jing Gao, Aiping Zheng

**Affiliations:** State Key Laboratory of Toxicology and Medical Countermeasures, Beijing Institute of Pharmacology and Toxicology, 27th Taiping Road, Haidian District, Beijing 100850, China; huayabing1111@126.com (Y.H.); wangzm.1986@163.com (Z.W.); wdgs223@163.com (D.W.); linxiaoming@luye.com (X.L.); 17326943829@163.com (B.L.); zhhui58@126.com (H.Z.)

**Keywords:** PLGA microspheres, reverse engineering, Vivitrol^®^, generic naltrexone-loaded microspheres, critical process parameters

## Abstract

The FDA (U.S. Food and Drug Administration) has approved only a negligible number of poly(lactide-co-glycolide) (PLGA)-based microsphere formulations, indicating the difficulty in developing a PLGA microsphere. A thorough understanding of microsphere formulations is essential to meet the challenge of developing innovative or generic microspheres. In this study, the key factors, especially the key process factors of the marketed PLGA microspheres, were revealed for the first time via a reverse engineering study on Vivitrol^®^ and verified by the development of a generic naltrexone-loaded microsphere (GNM). Qualitative and quantitative similarity with Vivitrol^®^, in terms of inactive ingredients, was accomplished by the determination of PLGA. Physicochemical characterization of Vivitrol^®^ helped to identify the critical process parameters in each manufacturing step. After being prepared according to the process parameters revealed by reverse engineering, the GNM demonstrated similarity to Vivitrol^®^ in terms of quality attributes and in vitro release (f_2_ = 65.3). The research on the development of bioequivalent microspheres based on the similar technology of Vivitrol^®^ will benefit the development of other generic or innovative microspheres.

## 1. Introduction

Long-acting injectable microspheres based on biodegradable poly(lactide-co-glycolide) (PLGA) have been used clinically since 1989. PLGA microspheres have drawn tremendous amounts of attention because of their availability, improved patient compliance, reduced administration frequency, and flattened blood concentration fluctuation. Despite their advantages and application history of more than 30 years [1], only 12 PLGA-based microsphere formulations have been approved by the FDA. This small number indicates the difficulty in developing a PLGA microsphere, and the consequent high cost. Based on a desire to promote the development of generic drugs to enable more patients to access affordable medicines, the FDA proposed the Drug Competition Action Plan [1] in June 2017. More than 2500 applications for generic drugs have been approved by the FDA from June 2017 to July 2020, but none of them were PLGA microspheres. The State Council of China launched a consistency evaluation for the quality and curative effect of generic drugs in March 2016, which also provided strong policy support for the development of generic drugs; however, no PLGA microsphere product has passed the consistency evaluation to date. A thorough understanding of microsphere formulations is essential to meet the challenge of developing generic microspheres.

Vivitrol^®^ is a PLGA microsphere product that encapsulates and slowly releases naltrexone (NTX) and is one of the only two approved small molecular drugs loaded with PLGA microspheres (alongside Risperdal Consta^®^). It can maintain stable and pharmacologically relevant plasma concentrations of NTX for at least 30 days. Vivitrol^®^ was approved by the FDA for the treatment of alcohol dependence in 2006 and opioid dependence in 2010 [2]. According to publications and packaging inserts, Vivitrol^®^ is supplied in single-use kits. Each kit contains a 380 mg vial of NTX microspheres and a vial containing 4.0 mL (to deliver 3.4 mL) of diluent to suspend the microspheres. The diluent for injection consists of sodium carboxymethyl cellulose (Na-CMC) 0.5% (*w*/*w*), mannitol 5% (*w*/*w*), Tween-20 0.1% (*w*/*w*), water and sodium chloride. Before administration, the microspheres are fully mixed with the diluent until a uniform suspension is formed. Due to the high drug loading and complexity of the manufacturing process of Vivitrol^®^, no generic drugs have been approved yet [3].

A generic medicine is the same as a brand-name medicine in terms of dosage, safety, effectiveness, strength, stability, quality, and the route of administration [4]. According to this standard, a generic product needs to be qualitatively (Q1) and quantitatively (Q2) the same as its listed reference drug, in terms of its inactive ingredients [5]. PLGA is the key ingredient of Vivitrol^®^, and thus, its composition and amount are of crucial importance for the release of the drug in microspheres [6]. Its identification is the first step in generic drug development. The systematic characterization of PLGA, such as lactic acid (LA)/glycolic acid ratio (GA) (LA/GA), molecular weight and distribution, and polymer end group, is critical for understanding and controlling drug-release kinetics from PLGA-based microspheres [7]. Gel permeation chromatography (GPC) was applied to measure the relative molecular weight of PLGA with external standards [8,9], and nuclear magnetic resonance (NMR) revealed the end-group of PLGA [8]. A combination of these techniques has provided a wealth of information on PLGA in Vivitrol^®^.

However, beyond Q1 and Q2, there are still many other parameters, especially in the manufacturing process, that affect the release profile of drugs from PLGA microparticles [10]. These parameters, such as the solvent and emulsifier used in preparation, the drug distribution in the microsphere, the apparent and intrinsic properties of the microspheres, and the in vitro testing method, also play important roles in the release behavior, and hence, the product performance of microspheres [11]. Minor changes in manufacturing processes can alter the physicochemical properties of Q1/Q2-equivalent microspheres [10,12]. The elucidation of the quality attributes and critical manufacturing process parameters of Vivitrol^®^ pose a great challenge in the development of bioequivalent generic drugs.

This study attempted to construct a bioequivalent generic naltrexone-loaded microsphere (GNM) based on similar technology to Vivitrol^®^ via reverse engineering study. Figure 1 shows that the critical process parameters and corresponding quality attributes of poly(lactide-co-glycolide) (PLGA) microspheres in the main manufacturing process. And the quality attributes could be obtained from the reverse engineering of Vivitrol^®^ and promoted the identification of key process parameters. The physicochemical properties and in vitro release of GNM were investigated to verify the speculation of manufacturing process. Q1 and Q2 similarity to Vivitrol^®^ in terms of inactive ingredients was accomplished via the determination of PLGA. Physicochemical characterization of Vivitrol^®^ helped to identify the critical process parameters in each manufacturing step. The in vitro release performance of the GNM was compared with that of Vivitrol^®^ to demonstrate the feasibility of such process parameters.

## 2. Results

### 2.1. Drug Loading

The drug loading analysis method was based on prior research [13,14,15,16,17], and has been verified in terms of through methodology (Appendix A). Based on Vivitrol^®^ packaging inserts, the microsphere powder was filled into transparent glass vials. There was 380 mg of naltrexone in each vial, contained in a biodegradable matrix of 75:25 polylactide-co-glycolide at a concentration of 337 mg of naltrexone per gram of microspheres, which indicates that the drug loading was approximately 33.70 wt% and that the content of PLGA was 66.30%. As shown in Table 1, Vivitrol^®^ and the GNM exhibited similar drug loading (33.50 ± 0.45 wt% and 34.62 ± 1.25 wt%, respectively) based on the description in packaging inserts and research by Andhariya, et al. [13]

### 2.2. Molecular Weight of PLGA in Vivitrol^®^

Three parameters were used to characterize the relative molecular weight of PLGA: namely, the weight-average molecular weight (Mw), number-average molecular weight (Mn), and polydispersity (PD, Mw/Mn). Relative molecular weight not only affects its own degradation rate, but also the release behavior of drugs contained in sustained- and controlled-release injectables; thus, it is essential to obtain PLGA’s relative molecular weight information for the study of Vivitrol^®^ microspheres. As shown in Table 1, the Mw, Mn, and PD of Vivitrol^®^ were 83118 ± 2698 Da, 47136 ± 1145 Da, and 1.89 ± 0.12, respectively. The Mw, Mn, and PD of the GNM were 81381 ± 2475 Da, 47376 ± 1480 Da, and 1.93 ± 0.16, respectively.

### 2.3. L/G Ratio and Content of PLGA in Vivitrol^®^

The L/G ratio is one of the most crucial attributes of PLGA regarding its ability to control long-acting release. Thus, even though it has been identified by the Vivitrol ^®^ drug label, the actual value was determined and verified carefully in this study. Because the GA bond is easily hydrolyzed, the L/G ratio in PLGA should be greater than 50:50 to ensure the longer sustained release of PLGA microspheres. As shown in Figure 2, the response peaks of dehydrated LA in PLGA appeared at approximately 1.5 and 5.2 ppm, respectively, corresponding to -CH_3_ and -CH groups in dehydrated LA. The response peaks of dehydrated GA appeared at 4.7 ppm, corresponding to the -CH2 group in dehydrated glycolic acid. The peak integration of the -CH- group in dehydrated LA was similar to that of dehydrated GA, and they were selected as the quantitative peaks of dehydrated LA and dehydrated GA, respectively. The peak of benzyl alcohol (BA) appeared at approximately 8.0 ppm [18]. The L/G ratios of the each of three batches of Vivitrol^®^ were 72: 28, 72.8: 27.2 and 72.7: 27.3, and the PLGA content was 66.82, 66.25 and 66.98%, respectively, which closely corresponded to the reported values of 75: 25 and 33.70 wt% of naltrexone and 66.30 wt% of PLGA in Vivitrol^®^.

### 2.4. End-Group of PLGA in Vivitrol^®^

The endcap analysis in this study was validated using PLGA with an acid end-cap and an ester end-cap, respectively. The results are shown in Figure 3. The only substantial difference between the two spectra was the presence of the methyl end-cap peak at 14 ppm (arrow in Figure 3) indicating the presence of an ester end-cap. There was a response peak at 14 ppm in PLGA with an ester end-cap, whereas there was no peak in PLGA with an acid end-cap. The ^13^C NMR spectra revealed that the PLGA in Vivitrol^®^ contains an ester end-cap.

### 2.5. Particle Size and Distribution

The particle size and distribution of Vivitrol^®^ and the GNM, which were detected by a laser-diffraction method and a microscopic analysis, are shown in Figure 4. The D_50_ of Vivitrol^®^ and the GNM—detected by the laser-diffraction method—were 79.33 ± 2.03 and 78.91 ± 3.08 μm, respectively, whereas the size of the same batches detected by optical microscopy were 55.52 ± 1.33 and 52.41 ± 3.52 μm, respectively. The particles in the local field of view were observed and measured under a microscope based on particle number, which was generally greater than 300. Laser-scattering particle-size determination is based on the scattering angle produced by the laser irradiation of particles; thus, the results reflect the particle volume, in which, the difference from the results of optical microscopy is explained. Although the results of these two methods differ, they are both methodologically validated and recognized for particle size detection (Appendix A).

### 2.6. Molecular Weight of PLGA in Organic Phase With Different Mixing Processes

As shown in Figure 5, the molecular weight of PLGA was determined to investigate the effects of different mixing procedures on the changes in PLGA chains. All PLGA dissolved in EA, with or without NTX, showed molecular weight losses to different extents.

### 2.7. Morphology

As shown in the scanning electron microscope (SEM) images, Vivitrol^®^ is spherical with some concavity and no apparent pores on the surface (Figure 6a,b). The internal structures in the Vivitrol^®^ and GNM cross-sections (Figure 6c,f) were relatively tight with some visible pores inside. This provides insight into the preparation process of Vivitrol ^®^.

### 2.8. Tg and Water Contact Angle

As shown in Table 1, the Tg of Vivitrol^®^ and the GNM was 47.32 ± 0.19 and 48.63 ± 3.73 °C, respectively (mean ± standard deviation (SD), *n* = 3), which provides guidance for the study of the method and mechanism of real-time and accelerated release in vitro [19]. The water contact angle of Vivitrol^®^ and the GNM was 64.41 ± 0.91 and 55.93 ± 0.38 °C, respectively, which may reflect the difference in a series of microsphere characteristics, such as porosity, particle size, and so on.

### 2.9. Residual EA

The residual EA of Vivitrol^®^ and the GNM determined by GC was 0.13 ± 0.03 and 0.19 ± 0.04%, respectively. For Vivitrol^®^, BA was added into the organic phase to promote the dissolution of NTX. The residual BA of Vivitrol^®^ and the GNM determined by HPLC was 0.07 ± 0.03 and 0.56 ± 0.06%, respectively. Moreover, no solvent residue of dichloromethane (DCM) was detected, which indicates that EA may be the main organic solvent in the preparation of Vivitrol^®^. This was further verified by SEM, which showed the different effects of EA and DCM on the morphology and internal structure of the microspheres. Generally, microspheres prepared by DCM have a rounder surface than those prepared by EA [20,21]. The porosity of Vivitrol^®^ and the GNM was 51.59 ± 2.56 and 59.11 ± 2.73%, respectively. However, SEM analysis showed that the surface of the particles appeared smooth without visible pores, which may be attributed to the pores on the surface being too small to be detected by SEM ^1^.

### 2.10. Raman Spectra

At 876, 1320, and 1758 cm^−1^ (Figure 7a), the PLGA signal at the shell was stronger than that at the core, which indicated that the PLGA tended to concentrate near the shell and not the core. This was responsible for the NTX distribution in the microspheres, and thus, was related to the release behavior. Figure 7b shows the drug distribution in Vivitrol^®^. The Ratio image shows a strong distribution of active pharmaceutical ingredient (API) in the corona and a weaker distribution at the edges. Raman spectra also showed that Vivitrol^®^ may have a different internal structure, which was consist with the results of the internal structure observed by cryo-SEM; this may be caused by drug loading and preparation processes.

### 2.11. Fourier Transform Infrared Spectra

As shown in Figure 8, the absorption peaks at approximately 1450, 1380, 1270, and 1190 cm^−1^ are due to -CH2 and -CH3 wagging and the deformation of PLGA [22]. The C=O bond of PLGA was absorbed at 1746 cm^−1^. The PLGA absorption peak was not shifted. There was some NTX absorption of Vivitrol^®^ at 1617, 1515, 633 and 591 cm^−1^. NTX exhibited some absorption at these wavelengths.

### 2.12. In Vitro Release and Degradation Studies

As shown in Figure 9, the cumulative release of Vivitrol^®^ lasted for 35 days with a very low initial burst release and almost no initial lag phase. More than 85% of naltrexone was dispersed by day 35 at 37 °C. The molecular weight profiles for microspheres in this study, normalized relative to the original Mw, along with the dispersity data are plotted in Figure 9. PLGA in Vivitrol^®^ decreased rapidly within 7 days, and there was no significant molecular weight loss after day 7. The polymer dispersity (PD) also decreased until day 17, and then gradually increased until it remained constant. Moreover, microspheres of lower Mw tended to aggregate, and their in vitro release may resemble that of a thin film. The pH gradually decreased in the first 7 days and then rapidly decreased until day 35. The final pH was about 6.85. The similarity factor (f_2_) is a parameter that measures the similarity of the dissolution curve of the two formulations, and f_2_ > 50 indicates similar dissolution behavior of the two preparations. The release characteristics of the GNM were similar to those of Vivitrol^®^ (f_2_ = 65.3).

Figure 10 shows Vivitrol^®^ morphology changes due to degradation. On the first day, a few microspheres cracked, while most of them remained intact (Figure 10,1d), which suggests that the degradation merely occurred in the beginning stages. From day 3 to day 5, more cracked microspheres appeared, showing that the degradation accelerated. From day 7 to day 21, more than half of the microspheres lost their primary structure, and those that remained were hollow inside but retained their smooth shell. From day 24 to day 35, the drug-released microspheres were broken up into pieces gradually, and the remaining shell surface shrank synchronously.

## 3. Discussion

The difficulty of generic microsphere development stems largely from the lack of a deep understanding of the original drug. A large amount of information, especially the critical process parameters, cannot be determined directly from the characteristics of the final microsphere products. The results of reverse-engineering studies should be considered comprehensively to disclose the quality attributes of PLGA microspheres and develop a feasible manufacturing process.

The equivalence of Q1 and Q2, that is, the type and amount of PLGA and NTX, represents the premise for developing bioequivalent microspheres. Drug loading and encapsulation efficiency were influenced by drug dispersity in the PLGA matrix, which was related to the PLGA composition (L/G ratio and the sequence of L and G), molecular weight, presence of PLGA end-caps (ester or carboxyl), and drug-polymer interactions [23], and these factors eventually affect the drug release kinetics as well as the degradation mechanism of the PLGA matrix [24,25]. The drug loading of Vivitrol^®^ was one of the highest (33.50 ± 0.45 wt%) among the marketed PLGA microspheres, which indicated that both the L/G ratio and molecular weight of the PLGA in it should be high enough to achieve a certain viscosity to facilitate the loading of more drugs and reduce the drug loss during manufacturing. The detected L/G ratio was 72.5:27.5, which is commonly used in PLGA drug-loaded microspheres. 

There is a close relationship between process parameters and key quality properties. The objective of the current study was to investigate the development of GNM by discovering the critical manufacturing parameters of NTX-loaded PLGA microspheres based on the analysis of quality attributes.

During the preparation of the organic phase, the sequence of dissolving PLGA and NTX in EA and BA affects the Mw change of PLGA. In particular, when PLGA and NTX were dissolved separately, PLGA showed a 20% decrease in Mw. The facilitation of the nucleophilic attack of NTX on PLGA may be responsible for the Mw decrease.

The Mw of PLGA deserves particular attention when selecting PLGA. The GNM was prepared using PLGA, of which, the Mw was 83,416 ± 1557 Da because the detected Mw of PLGA in Vivitrol^®^ was about 80,000 Da. However, the Mw in the resultant microspheres was only 47695 ± 273 Da (Appendix A). When the Mw of PLGA used in preparation was increased to 140928 ± 774 Da, the detected Mw of PLGA in the GNM became similar to that of Vivitrol^®^ (81381 ± 2475 Da), which means that 42.25% of the molecular weight loss occurred during the preparation. The loss of PLGA molecular weight may be attributed to the hydrolysis of ester bonds when PLGA is mixed with nucleophilic drugs, such as NTX and risperidone in solvents to prepare an organic phase [14].

Tg is the point at which PLGA changes from an amorphous state to a highly-elastic state. The Tg value represents the block length and arrangement of the lactic and glycolic acid chains as well as the polymer molecular weight of PLGA. As one of the potential key factors that determine release behavior, the Tg of PLGA also needs to be considered. A different Tg may indicate a different drug release mechanism [26]. The Tg of the microspheres could be mainly affected by the Tg of the raw polymer and the interaction between NTX and PLGA chains. Okada et al. found that the Tg of drug-loaded microspheres increased gradually from 42 to 47 °C as drug loading increased [27]. The Tg of Vivitrol^®^ was determined to be 47.32 ± 0.19 °C. When the temperature is higher than the Tg, degradation and drug release tend to accelerate in the polymer in a highly-elastic state [28,29]. The loaded drug and excess emulsifier (such as polyvinyl alcohol, PVA) residue may accelerate the degradation of microspheres owning to the plasticizing effect of the drug and the emulsifier, which is followed by a decrease in Tg [30].

EA and DCM are the solvents most frequently used to dissolve PLGA, especially when the solvent evaporation method is applied. The results of the residual solvents analysis showed that EA was employed in the preparation of Vivitrol^®^. Furthermore, EA and water are partially miscible; thus, EA and water move dynamically during microsphere solidification [19,31]. This dynamic movement leads to water inclusion in PLGA microspheres, resulting in an irregular shape and indentation during the drying process [17]. The irregular morphology of Vivitrol^®^ indicates that EA may be used as an organic solvent in microsphere manufacturing. To find further evidence of EA as the preferred solvent, GNMs were also prepared using DCM and observed by SEM (Appendix A), they showed significant differences in morphology compared with those prepared using EA. Microspheres prepared using DCM seemed relatively round, lacked concavity, were porous on the surface, and had apparent differences with Vivitrol^®^.

Particle size and distribution are the quality attributes of microspheres, which are equally as important as drug loading. These attributes play a vital role in the sustained release performance and guarantee needle penetration. A high PLGA concentration always results in high drug loading, but a small particle size. The narrow particle-size distribution also requires an appropriate PLGA concentration. The concentration of PLGA was investigated with drug loading, particle size and distribution, and Mw as evaluation indices (which were similar to those of Vivitrol^®^) and 16.7% *w*/*w* was found to be the appropriate concentration of PLGA.

A small amount of a drug can be easily encapsulated when it is dissolved in an appropriate solvent with PLGA to form true or metastable molecular dispersions [25]. However, NTX is relatively poorly soluble in EA, and the drug loading of Vivitrol^®^ is high; thus, a co-solvent should be used to facilitate the dissolution of NTX. Residual BA was also measured by HPLC, which showed that BA acted as a co-solvent to dissolve NTX in the preparation of the organic phase.

Drug loading is also closely related to particle size, which may result from the final surface/volume ratio of the PLGA microspheres and discrepancies in the rates of solvent extraction between small and large emulsion droplets during preparation [32]. Particle size and distribution are a comprehensive result of various formulation and process parameters, but they mainly depend on the PLGA concentration and the emulsifying shear force. High-speed homogenization is widely utilized in the continuous large-scale production process of marketed microsphere products. The resultant microsphere has a relatively wide particle distribution, which is not conducive to batch-to-batch reproducibility [33]. Vivitrol^®^ has a narrow particle-size distribution, which indicates that the microspheres might have been sieved to control their size after solidification or before filling. Through these parameters, it was found that a homogeneous rotation speed of 3000 rpm resulted in a GNM with a similar particle size and distribution as Vivitrol^®^.

During solidification, the formation of pores and the change in morphology of the microspheres should be considered. Some concavity appeared on the surface of Vivitrol^®^ and the GNM, and pores were apparent in the interior of Vivitrol^®^ and the GNM. These pores may be attributable to the fact that the solvent was exchanged with water during the emulsifying evaporation, and then the microspheres were shrunk under vacuum. An aqueous solution containing 2.5% EA was used as the medium of solidification, and then microspheres formed under vacuum, which exhibited similar morphology and internal structure to those of Vivitrol^®^. Before the bulk erosion of PLGA takes place, sufficient microsphere porosity must be generated to facilitate drug diffusion and subsequent release [34]. Higher porosity facilitates faster drug release. Changes in the manufacturing process (such as different solvents, and different rates of solvent diffusion and evaporation) have been reported to affect the inner structure and/or porosity of PLGA microspheres. It has been reported that rapid solvent removal and polymer precipitation lead to the formation of large pores but low porosity [20,28]. Generally, substances with lower boiling points have faster volatilization rates. The boiling point of DCM is 39.75 °C, while that of EA is 77 °C. Compared with the porosity of GNMs using DCM (47.29%, similar to the 49.83% reported by Burgess et al.13), higher porosity was observed in Vivitrol^®^ and the GNM when EA was used as the solvent. There was almost no lag phase in the release of Vivitrol^®^ due to the high porosity.

Drug release from PLGA microspheres could be explained by diffusion through water-filled pores, diffusion through the PLGA, osmotic pumping, erosion, and hydrolysis [28,29,35]. Regarding the release properties of Vivitrol^®^, the initial release stage occurred on the first day after the microsphere was exposed to an aqueous environment, which releases some drugs on or near the surface. In this stage, a high-burst release should be avoided. The hydration stage was observed in the first week, wherein the physical erosion of the microspheres began, and some subsurface drugs were released. In this stage, continuous exposure was provided, and 30% of NTX was released from the microsphere. Subsequently, the continuous release phase began from the second week until drug was completely released. The release of microspheres was controlled by polymer erosion. The objective of this stage was to release the remaining encapsulated drugs at a steady rate. Vivitrol^®^ and the GNM showed bi-phasic release profiles with a small burst release phase followed by continuous zero-order release over 35 days. These in vitro release profiles could depend on the physicochemical properties of PLGA (such as L/G ratio, molecular weight, crystallinity, and monomer sequence) and the drug, as well as the critical quality attributes of microspheres (such as drug loading, particle size, morphology, porosity, Tg, and contact angle) [19]. These data are consistent with the published literature, which shows that Vivitrol^®^ does not have a lag phase in vitro, potentially owing to enhanced polymer degradation as a result of quickly decreasing local acidic pH after one week, hydrolysis, and these accelerated PLGA degradations [26,27]. The PLGA ester bonds containing more GA are less stable than the bonds with more LA, as rapid cleavage of G–G linkages would be expected to result in earlier increases in polymer chain dispersity [30]. As the release process depends on the degradation of PLGA ester bonds, the composition of monomers changes over time during release, typically resulting in an increase the in L/G ratio.

The results of Raman spectroscopy showed that most of the drug was distributed in the corona of the microsphere instead of the surface, which can explain the release behavior of the microspheres. The initial release phase of the microspheres occurs on the first day after exposure to an aqueous environment, and the burst release does not appear because the little drug is kept at or near the surface. The hydration phase occurs during the first week, and the drug in the matrix begins to be released. From the second week onwards, the drug near the corona is released in a sustained manner accompanied by polymer erosion. Each step of the manufacturing process may affect the drug distribution in the microspheres. Particularly in solidification, NTX prefers to stay with PLGA rather than leave with solvents from the inner microspheres.

The water contact angle was used to determine the wettability of the microspheres, an index reflecting the water absorption rate at the beginning of drug release [36]. The microspheres were pressed into a thin tablet before the contact angle was determined (Appendix A). The detected water contact angle of the GNM was lower than that of Vivitrol^®^. The larger the water contact angle, the more hydrophobic the microsphere will be, which indicates that the polymer chain may be a hydrophobic diffusion barrier [37]. The contact angles of the GNM and Vivitrol^®^ are somewhat different. This may be due to the wider distribution of the GNM compared with that of Vivitrol^®^. Alternatively, this may result from the slight difference in the PLGA used in the GNM and Vivitrol^®^. In the same manner as porosity, drug release is accelerated if the microsphere tends to be more hydrophilic.

The final process of lyophilization may be related to the moisture content of the microspheres. This indicated that the moisture content of the GNM is consistent with that of Vivitrol^®^.

The above results indicate that minor changes in the manufacturing process may give rise to different drug loading characteristics porosities, and drug distribution, thus leading to changes in release behavior. Because it was prepared according to the process parameters revealed by reverse engineering, the quality attributes and in vitro release of the GNM were similar to those of Vivitrol^®^. The key process factors of the marketed PLGA microspheres were revealed for the first time via a reverse engineering study on Vivitrol^®^ and a GNM, as prepared according to these key process parameters. It was found that most of the quality attributes of the GNM were similar to those of Vivitrol^®^. Furthermore, its release behavior in vitro was similar to that of Vivitrol^®^. However, the BA residue and porosity of the GNM were relatively high. The high residue of BA may be attributed to the further removal of organic solvents by other measures (such as extraction) after solidification. Similarly, the difference in the control parameters of the solvent-evaporation rate during solidification of the microspheres may lead to the high porosity of the GNM. The lower contact angle indicates that this is due to the slight difference in particle-size distribution. Alternatively, it may be attributed to the fact that although the proportion of LA/GA ratio, terminal group, and molecular weight of the selected PLGA and Vivitrol^®^ are similar, the PLGA produced by different manufacturers and different processes will still be quite different. Regarding the similar in vitro release behavior of Vivitrol^®^ and the GNM, it seems that the porosity and contact angle of NTX microspheres have little effect on their release.

## 4. Materials and Methods

### 4.1. Materials

Vivitrol^®^ was purchased from Alkermes, Inc. PLGA was obtained from Merck (Darmstadt, Germany). Anhydrous naltrexone base was purchased from Chongqing Land Tower (Chongqing, China). Ethyl acetate (EA) was purchased from Nanjing Chemical Reagent Co., Ltd. (Nanjing, China). Benzyl alcohol (BA) was purchased from Shanghai Aladdin Biochemical Technology Co., Ltd (Shanghai, China). Dimethyl sulfoxide (DMSO) was purchased from Thermo Fisher Scientific (Waltham, MA, USA). Milli-Q^®^ water was used for all the studies. All other solvents used were of HPLC grade and purchased from Thermo Fisher Scientific.

### 4.2. Characterization of Q1/Q2 PLGA Microspheres

#### 4.2.1. High Performance Liquid Chromatography (HPLC)

The quantification of naltrexone in Vivitrol^®^ was conducted using the 1260 Infinity HPLC system (Agilent, Santa Clara, CA, USA) with a UV absorbance detector set to 210 nm. The mobile phase was 10 mM phosphate buffer (pH 6.6)/methanol (35/65, *v/v*), and the flow rate was 1 mL/min. A polar C18 column (250 × 4.6 mm, 5 μm; SHISEIDO Technologies) was used as the stationary phase. The sample injection volume was 10 μL for drug-loading detection [13,14].

#### 4.2.2. Drug Loading

The microspheres (~47 mg) were weighed, dissolved in 10 mL of DMSO and diluted 10 times with methanol. The solutions were filtered through a 0.22 μm polyvinylidene fluoride (PVDF) syringe filter and detected using an HPLC assay, as described in the section “High performance liquid chromatography (HPLC)”. The drug loading of Vivitrol^®^ was calculated using the following Equation (1):(1)$$\mathrm{Drug}\;\mathrm{Loading}\;(\%)\;=\frac{weight\;of\;drug\;entrapped}{weight\;of\;microspheres\;analyzed}\;\times 100$$

#### 4.2.3. The Molecular Weight of PLGA

The molecular weight was determined via gel-permeation chromatography (GPC) (Thermo Fisher, USA) equipped with a Styragel guard column (4.6 × 30 mm, Waters, Milford, MA, USA), two Styragel columns (HR 4 and HR 4E columns, 7.8 × 300 mm, Waters, Milford, MA, USA) and a refractive index detector (RefractoMax520, IDEX Health & Science KK, Shanghai, China). The PLGA sample was prepared as follows: Microspheres (~10 mg) were weighed and dissolved in 10 mL tetrahydrofuran (THF). The PLGA was fully dissolved when it was stored at room temperature for more than 12 h. The sample was filtered with 0.22 μm PVDF. Polystyrene standards with a weight-average molecular weight (Mw) ranging from 4000 to 150,000 Da were dissolved in dehydrated THF. The Mw and PD (polydispersity) of the PLGA were calculated using Chameleon software (Thermo, Fisher, USA).

#### 4.2.4. Analysis of Content and Lactide/Glycolide (L/G) Ratio of PLGA in Vivitrol^®^

The content and L/G molar ratio of PLGA were determined by ^1^H NMR spectroscopy (Bruker Fourier 300, Germany). Vivitrol^®^, PLGA, and benzoic acid (BA, internal standard) were dissolved in 1 mL deuterated chloroform (CDCl_3_) and pipetted into an NMR tube for collection. Instrument parameters were as follows: 90° pulse, relaxation delay time D1 = 20 s, acquisition time (AQ) = 4.4 s, temperature = 298.0 K, sampling data points (TD) = 64 K, spectral width = 15 ppm, scanning times (NS) = 16 times. The L/G was determined from the proton signals generated by -CH groups of lactide (LA) at 5.2 ppm and -CH_2_ groups of glycolide (GA) at 4.8 ppm using Equation (2) and the content of PLGA were calculated using Equation (3).
(2)$$\frac{L}{G}=\frac{2PI_{LA}}{2PI_{LA}+PI_{LG}}\;:\frac{PI_{LG}}{2PI_{LA}+PI_{LG}}$$
(3)$$M_{PLGA}=\;\frac{M_{BA}{\text{ × }P}_{\mathrm{BA}}{\text{ × }(\mathrm{PI}}_{\mathrm{LA}}{\text{ × }2\text{ × }\mathrm{Mw}}_{\mathrm{LA}}+{\mathrm{PI}}_{\mathrm{GA}\;}{\text{× }\mathrm{Mw}}_{\mathrm{GA}})}{{\mathrm{Mw}}_{\mathrm{BA}}{\text{ × }\mathrm{PI}}_{\mathrm{BA}}}$$

“M_PLGA_” and “M_BA_” represent the mass of PLGA and BA, respectively; “PI_LA_”, “PI_GA_” and “PI_BA_” are the peak integration of LA, GA, and BA, respectively. “Mw_LA_”, “Mw_GA_” and “Mw_BA_” represent the Mw of LA, GA, and BA, respectively. And “P_BA_” is the purity of BA, the purity of PLGA was considered as 100%.

#### 4.2.5. End-Group Analysis of PLGA in Vivitrol^®^

The end-group of PLGA was investigated by ^13^C NMR [8]. PLGA was dissolved in CDCl3 and transferred into NMR tubes to collect ^13^C NMR at 600 MHz. In the test, a Z-restored spin–echo pulse sequence was utilized with a 30-degree observation pulse, a 3 s interpulse delay, and a 0.55 s data acquisition time. A total of 12,000 scans were acquired over 12.5 h. The data were processed with exponential multiplication (line-broadening factor of 3) and baseline straightening prior to plotting. The presence of an ester end-cap was determined by the existence of a peak at ~14 ppm.

### 4.3. Physicochemical Characterization for Critical Process Parameters

#### 4.3.1. Particle Size and Distribution

The particle size and distribution of the microspheres were measured using a Malvern Mastersizer 2000 (Malvern Instruments Ltd., Worcestershire, UK) (optical mode and refractive index). Briefly, approximately 300 mg of microspheres was suspended in 500 µL of 1.0% Tween-20 and vortexed vigorously before being adding them the instrument sample dispersion unit, and particle size analysis was conducted. Three measurements were performed per sample at a stirring speed of 2100 rpm and a sampling time of 15 s. At the same time, the particle size of Vivitrol^®^ was calculated using a scanning electron microscope (SEM) and Nano Measurer.

#### 4.3.2. Morphology and Internal Structure

The surface morphology and internal structure of the microspheres were examined using a Hitachi S3200N scanning electron microscope (SEM) (Hitachi, Tokyo, Japan). The samples were fixed on a brass stub using double-sided carbon adhesive tape and prepared to be electrically conductive by coating with a thin layer of gold for 120 s at 40 W under vacuum conditions. Images were taken at an excitation voltage of 10.0 kV. To examine the internal structure of the microspheres, they were placed in a −80 °C freezer before being transferred into the SEM. The microspheres were quickly removed from the freezer and then cross-sectioned using a blade to study the internal microstructure, which prevented deformation of the microspheres during cross sectioning due to their glass-like structure.

#### 4.3.3. Glass Transition Temperature

The glass transition temperature (Tg) of microspheres was determined using a differential scanning calorimeter (DSC Q2000, TA Instruments, New Castle, USA). Microspheres (~5 mg) were crimped in DSC aluminum pans. Temperatures were ramped between 0 and 190 °C at a rate of 5 °C/min. The samples were subjected to a heat/cool/heat cycle. The results were analyzed using Origin 8.0 software, and Tg was taken at the midpoint of the revers heating event, *n* = 3.

#### 4.3.4. Residual Solvent

The residual solvent in the microspheres was determined by a Trace 1310 gas chromatograph (GC) with a hydrogen flame ionization detector (FID) (Thermo Fisher Scientific Inc., Waltham, MA, USA). The microspheres (~500 mg) were added to 10 mL DMSO in volumetric flasks to obtain a sample for the test. EA was added to a volumetric flask containing DMSO to yield a final concentration of ~250 μg/mL, which was used as a reference for the test. The chromatographic column was DB-624 (30 mm, 0.32 mm × 1.8 μm). The initial temperature was maintained at 40 °C for 10 min, increased to 160 °C at a rate of 10 °C /min, increased to 220 °C at a rate of 10 °C/min and then maintained 220 °C for 5 min, until the detector temperature increased to 250 °C. The residual percentage of EA was calculated using the peak area according to the external standard method and should not exceed 0.5%.

The residual BA was detected using the method described in the section “High performance liquid chromatography (HPLC)”. Briefly, 10 µL of BA was added to 25 mL acetonitrile and was used as a control solution. Approximately 13 mg of microspheres was added to 25 mL acetonitrile and used as a test solution.

#### 4.3.5. Porosity

The porosity of the microspheres was determined using a mercury porosimeter (Poremaster GT60, Anton Paar, Graz, Austrian). Briefly, approximately 200 mg of microspheres was introduced into the porosimeter and tested at a mercury-filling pressure of 0.53 psi. The total intrusion volume, total pore area and porosity (%) were recorded. (porosity (%) = bulk density/apparent (skeletal) density × 100%).

#### 4.3.6. Water Contact Angle

Water contact angle measurements were recorded using a VCA optima XE video (KRUSS) contact angle system at 25 °C and 42–48% relative humidity. Approximately 200 mg of microspheres was pressed into tablets. A droplet was formed at the end of the needle and then lowered carefully until contact was made with the sample. The needle was withdrawn immediately so that the droplet was left on the sample surface. An image of the droplet was acquired with a charge-coupled device (CCD) camera 1.7 s after contact with the surface of the sample. The static contact angle was calculated automatically using the VCA software. Approximately 30 s was required to complete the entire measurement process.

#### 4.3.7. Residual Moisture

Briefly, approximately 0.5 g of microspheres was placed on a rapid moisture tester (MJ33, Mettler) tray for moisture determination, and the residual moisture of the microspheres was recorded after 5 min.

#### 4.3.8. Raman Spectroscopy

The measurements were performed on a DXR2xi microscopic imaging Raman spectrometer (Thermo Fisher). The spectra of Vivitrol^®^, NTX and PLGA were obtained. Data acquisition and analysis were performed using OMNIC Version 9.2 software.

Raman imaging measurements (drug distribution) were performed via RA802 micro-Raman spectroscopic analysis (Renishaw, Wotton-under-Edge, United Kingdom). The spectra of Vivitrol^®^, NTX and PLGA were obtained. The excitation wavelength was 785 nm, and the step size was 1 μm.

#### 4.3.9. Fourier Transform Infrared Spectra

Fourier transform infrared (FT-IR) spectra for Vivitrol^®^, NTX and PLGA were collected using a Fourier transform infrared spectrometer (Nicolet 50, Thermo Fisher) with a spectral resolution of 4 cm^−1^. The IR spectra in the wavenumber range of 400–4000 cm^−1^ were recorded for further comparison.

### 4.4. Preparation of Naltrexone Loaded PLGA Microspheres

Briefly, PLGA (2 g) was dissolved in EA (10 g). Naltrexone (1.336 g) was dissolved in benzyl alcohol (BA, a co-solvent, 3.12 g). The PLGA solution was added to the naltrexone solution and mixed well to form the organic phase. The organic phase was then dispersed into a 1% (*w/v*) PVA solution (40 mL, 0.22 μm membrane filtered) and homogenized (homogenization, IKA^®^ Works, Inc., Wilmington, NC, USA) at 3000 rpm for 8 min. The O/W emulsion was added to water (1200 mL) and stirred at 130 rpm for 6 h to allow microsphere solidification. The solvents were removed under vacuum at room temperature. The naltrexone microspheres were collected, washed with water, and dried after lyophilization. The microspheres were sieved using two sieves, including a 125 μm sieve on the top, and a 25 μm sieve on the bottom.

### 4.5. In Vitro Release and Degradation Studies

In vitro release testing of Vivitrol^®^ and the GNM was conducted using a sample-and-separate method. Briefly, 20 mg of microspheres was suspended in a 50 mL centrifugal tube with 50 mL phosphate buffered saline (PBS, 10 mM, pH 7.4) with 0.02% (*v/v*) Tween-20 and 0.02% (*w/v*) sodium azide and incubated in a water shaker bath at 100 rpm and at 37 °C. At pre-determined time intervals, 1 mL of the release sample was withdrawn and replenished with fresh medium. The release medium was replaced once per week. Medium replacement during release testing was considered in the calculation of the fraction release. All drug release tests were conducted in triplicate; the results are reported as the mean cumulative release (%) ± SD, and the f_2_ factor was calculated (f_2_ = 50 × Log [(∑(Rt−Tt)^2^/n) + 1]^−0.5^ × 100). It should be noted that at pre-determined time intervals, another certain microsphere (*n* = 3) was taken, freeze-dried directly, observed by SEM, and dissolved in THF to analyze the molecular weight via GPC. In addition, the pH of the release medium was continuously monitored.

### 4.6. Statistical Analysis

All values are expressed as the mean ± standard deviation (SD). Significant differences were calculated by a paired Student’s t-test, and a *p*-value < 0.05 was considered statistically significant.

## 5. Conclusions

The present study addresses key factors in the development of generic PLGA microspheres based on the composition, characteristics, release profile and in vitro behavior of a marketed long-acting microsphere product, Vivitrol^®^. A detailed physicochemical characterization of the PLGA microspheres was followed by a critical analysis and interpretation of the experimental data. These data facilitated the development of a bioequivalent GNM, based on similar technology as that involved in the manufacturing process of Vivitrol^®^. These finding will benefit the development of bioequivalent devices and innovative microspheres loaded with other small molecular drugs.

## Figures and Tables

**Figure 1 molecules-26-01247-f001:**
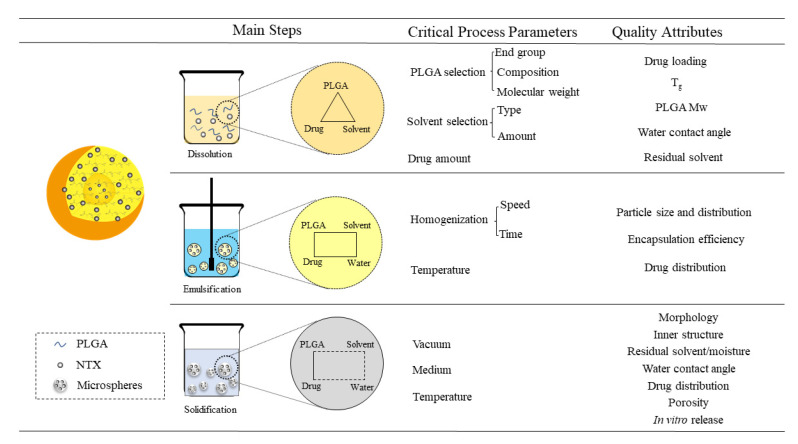
The critical process parameters and corresponding quality attributes of poly(lactide-co-glycolide) (PLGA) microspheres in the main manufacturing process. The quality attributes were obtained from the reverse engineering of Vivitrol^®^, and promoted the identification of key process parameters. NTX: naltrexone, Mw: the weight-average molecular weight.

**Figure 2 molecules-26-01247-f002:**
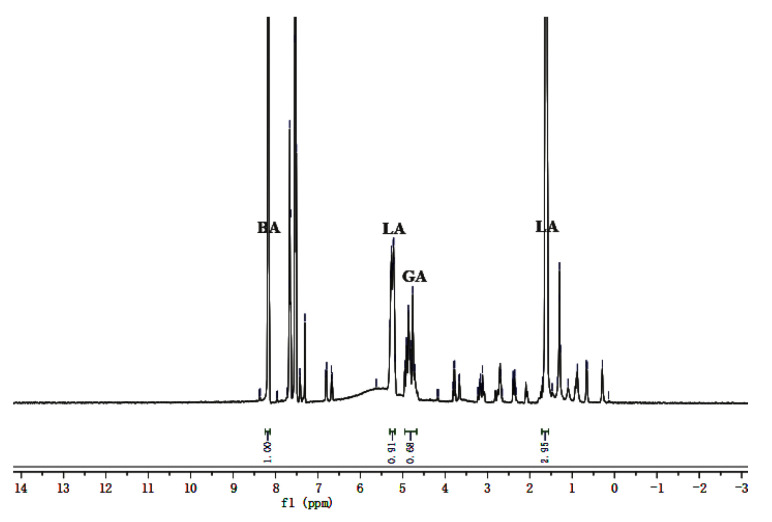
The ^1^H NMR of Vivitrol^®^. The response peaks of LA in PLGA appeared at about 1.5 and 5.2 ppm, corresponding to the -CH_3_ and -CH groups. The response peaks of GA appeared at 4.7 ppm, corresponding to the -CH_2_ group. LA: lactic acid, GA: glycolic acid, BA: Benzyl alcohol.

**Figure 3 molecules-26-01247-f003:**
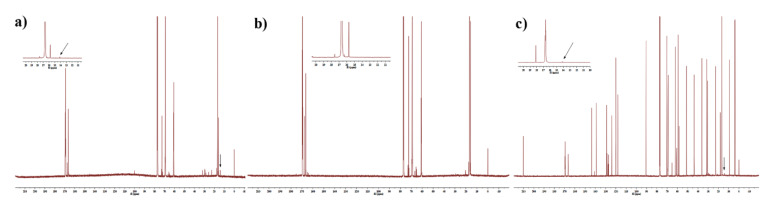
The ^13C^ NMR of Vivitrol^®^. There was a response peak at 14 ppm in PLGA with an ester end-cap (**a**) and Vivitrol^®^ (**b**), whereas there was none for the PLGA with an acid end-cap (**c**).

**Figure 4 molecules-26-01247-f004:**
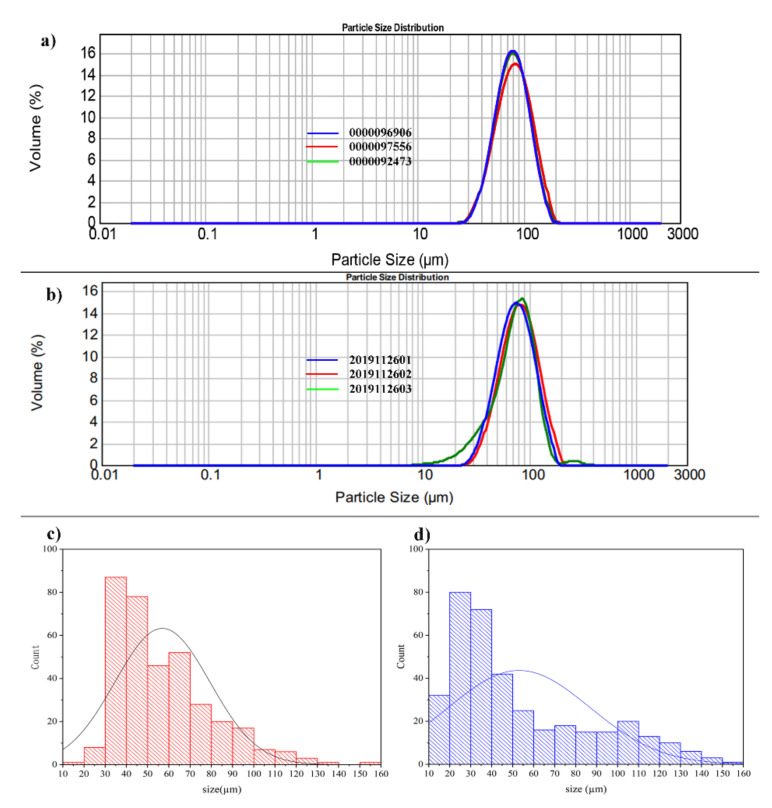
Particle size distribution of Vivitrol^®^ and the GNM. The laser-diffraction (**a**) and (**b**) and the microscope method (**c**) and (**d**) were both used to detect the particle-size distribution of Vivitrol^®^ and the GNM.

**Figure 5 molecules-26-01247-f005:**
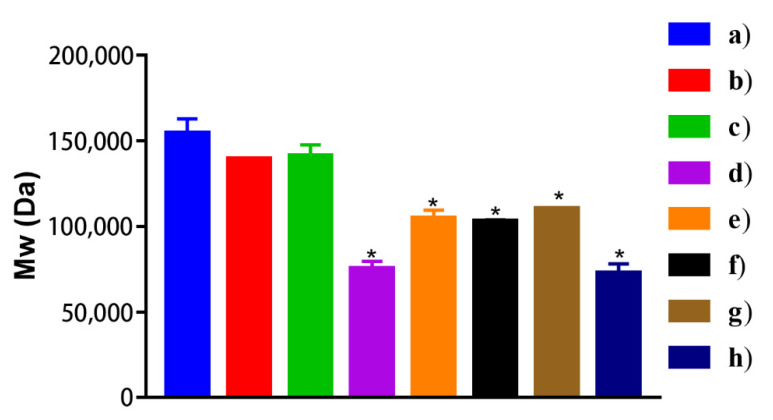
Mw of organic phase with different mixing procedures. Group a: only PLGA; group b: PLGA was dissolved in EA; group c: PLGA was dissolved PLGA in EA and BA; group d: PLGA and NTX were dissolved together in the mixture of EA and BA; group e: PLGA was dissolved in EA, followed by the addition of NTX, stirring for 5 min, and finally the addition of BA; group f: PLGA was dissolved in EA, followed by the addition of BA, stirring for 5 min, and finally, the addition of NTX; group g: NTX was dissolved in BA, followed by the addition of EA, stirring for 5 min, and finally, the addition of PLGA; group h: PLGA was dissolved in EA, NTX was dissolved in BA, and the two solution were mixed. EA: ethyl acetate, BA: benzyl alcohol. Compared with group a, **p* < 0.05.

**Figure 6 molecules-26-01247-f006:**
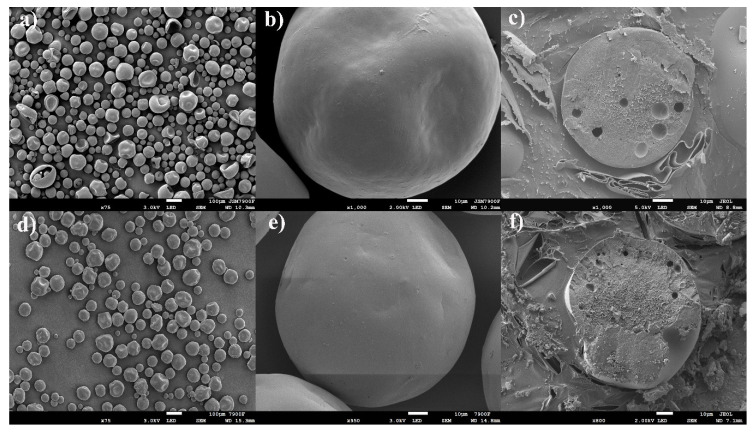
The morphology and relatively tight internal structure, Vivitrol^®^: (**a**–**c**); GNM: (**d**–**f**).

**Figure 7 molecules-26-01247-f007:**
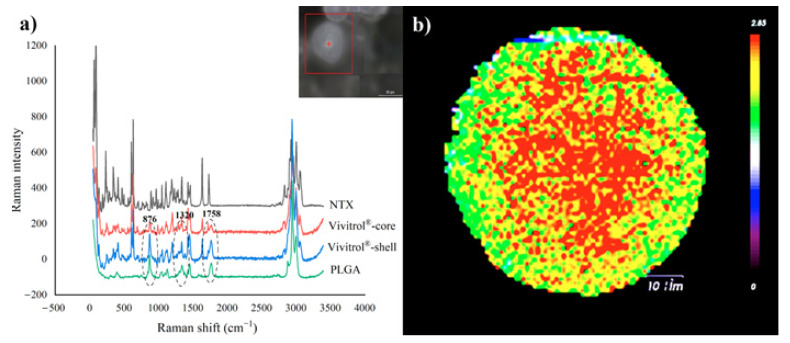
(**a**): Raman shift of NTX, Vivitrol^®^ (core and shell) and PLGA, (**b**): Drug distribution image in Vivitrol^®.^

**Figure 8 molecules-26-01247-f008:**
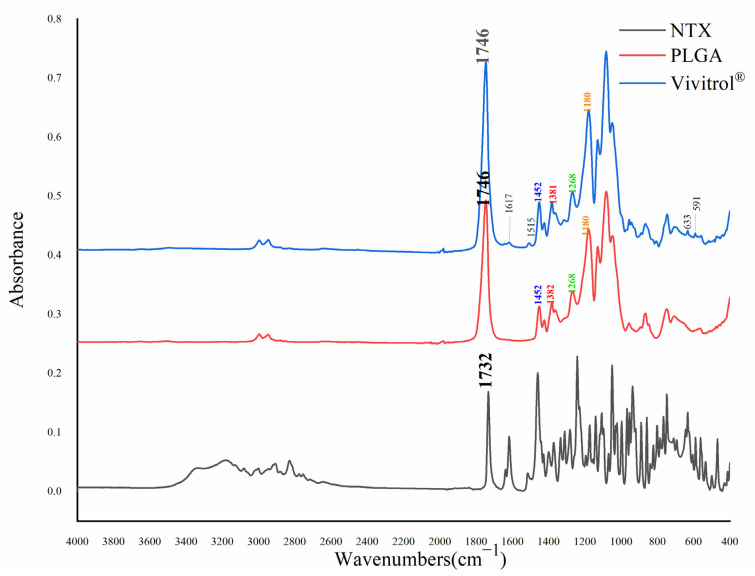
Infra-red spectra of NTX, PLGA and Vivitrol^®.^

**Figure 9 molecules-26-01247-f009:**
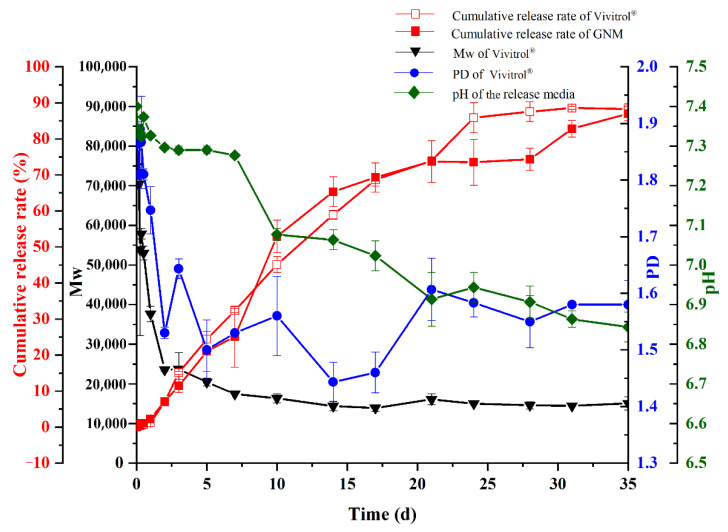
In vitro release of Vivitrol^®^ and GNM.

**Figure 10 molecules-26-01247-f010:**
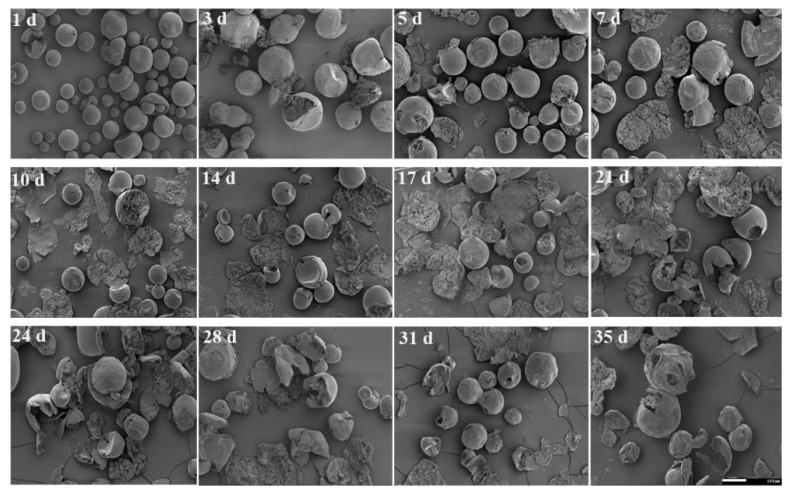
Degradation of Vivitrol^®^ during in vitro release observed by SEM.

**Table 1 molecules-26-01247-t001:** Characteristics of Vivitrol^®^ and a generic naltrexone-loaded microsphere (GNM).

	Vivitrol^®^	GNM
Batch No.	0000097556 0000092473 0000096906	2019112601
		2019112602
		2019112603
Drug loading (%)	33.50 ± 0.45	34.62 ± 1.25
Molecular weight (Da)/ distribution		
Mw	83118 ± 2698	81381 ± 2475
Mn	47136 ± 1145	47376 ± 1480
PD	1.89 ± 0.12	1.93 ± 0.16
Tg (ºC)	47.32 ± 0.19	48.63 ± 3.73
Residual solvent (%)		
EA	0.13 ± 0.03	0.19 ± 0.04
BA	0.07 ± 0.03	0.56 ± 0.06
Porosity		
Average pore diameter(µm)	0.138 ± 0.006	0.194 ± 0.063
Porosity (%)	51.59 ± 2.56	59.109 ± 2.73
Water contact angle (º)	64.41 ± 0.91	55.93 ± 0.38
Residual moisture (%)	2.32 ± 0.13	2.67 ± 0.42

Mw: the weight-average molecular, Mn: Number-average molecular weight, PD: polydispersity, EA: ethyl acetate, BA: benzyl alcohol.

## Data Availability

The data presented in this study are available in insert article or supplementary material here.

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
