# Peer review of "Key Factor Study for Generic Long-Acting PLGA Microspheres Based on a Reverse Engineering of Vivitrol®"

_molecules, 2021, doi:10.3390/molecules26051247_

Round 1

Reviewer 1 Report

The study of Hua et al appears well made and it does not seem to need technical modification. For that reason, I consider this manuscript publishable after a minor revision, following the suggestions reported.  

First of all, the manuscript requires a strong revision of writing in the English language and formatting style (i.e. see well the Abstract).

In my opinion, the Discussion section has to be improved: I suggest to eliminate the reference to the figure of the manuscript in order to make more fluid the text.

Therefore, the most important thing is about the main aim of the whole work. The description of “the manufacturing process of Vivitrol®  in as much detail as possible” does not appear as a so important issue. I suggest to change the point of view addressing the research to the development of bioequivalent devices based on the similar technology of the Vivitrol®. For that reason, some part of the text should be revised reformatting the starting hypothesis and the aim.

Reviewer 2 Report

This manuscript described the study of the key factor for generic long-acting PLGA microspheres based on a reverse engineering of Vivitrol. Overall, the preparation and characterization seem to be done properly, and the results have been reasonably interpreted. Therefore, I recommended the publication of this study in Molecules.

Reviewer 3 Report

This study addressed the key factors to develop generic PLGA microspheres based on the composition, characteristics, release profile and in vitro of marketed long-acting microspheres Vivitorol®. The reverse engineering study of marketed medicine might be benefit the development of generic microsheres in case of long-acting type especially. Each parameter used for evaluation to find the key factors is important and appropriate as a formulation preparation methods. However, the novelty about the results obtained is unclear to exhibit useful knowledge for development of formulation technology.

Is it clear that there is a difference in physical properties between Vivitorol® and GNM, but is there a correlation between the difference and the drug efficacy? Furthermore, it is unclear what should be improved when the difference in physical properties is clarified in Vivitorol® and GNM. Positive controls are unclear from the perspective of research for pharmaceutical development. That is why, I can not accept this article to publish in this journal.

Round 2

Reviewer 3 Report

The manuscript has been significantly improved.

In the original text, the purpose of this study was unclear. Furthermore, it was unclear for reader to understand the novelty of the results although the experimental items are detail. However, it became clear these issues by adding the following sentences in revised text in page 12.

Its release behavior in vitro was similar to that of Vivitrol. However, the BA residue and porosity of the GNM were relatively high.

It is very important to solve the above questions and consider the causes. And it provides important insights for development of new formulation in the future.

As you suggested in the response to reviewer comments, I expect that the solvent residue and internal microstructure will be thoroughly researched, including the drug efficacy they may bring for the generic microsphere. Impurity content has a large effect on BA when the polymer material or structure of cage is different even if the release profile in vitro was similar to original medicine such as Vivitrol.

The novelty of this study has been revealed by author’s appropriate corrections. This paper deserves to be published in this journal. I can accept that this article will be published in this journal.